# Lyso-Gb3 Increases αvβ3 Integrin Gene Expression in Cultured Human Podocytes in Fabry Nephropathy

**DOI:** 10.3390/jcm9113659

**Published:** 2020-11-13

**Authors:** Hernán Trimarchi, Alberto Ortiz, Maria Dolores Sánchez-Niño

**Affiliations:** 1Nephrology Service, Hospital Británico de Buenos Aires, 1280 Buenos Aires, Argentina; 2IIS-Fundación Jimenez Díaz, School of Medicine, UAM, 28040 Madrid, Spain; aortiz@fjd.es (A.O.); mdsanchez@fjd.es (M.D.S.-N.); 3Spanish Renal Research Network (REDINREN), 28004 Madrid, Spain; 4Pharmacology and Therapeutics Department, Universidad Autonoma de Madrid, 28049 Madrid, Spain

**Keywords:** Fabry disease, lyso-Gb3, podocyte, podocyturia, αvβ3 integrin, proteinuria

## Abstract

Background: Podocyturia in Fabry nephropathy leads to glomerulosclerosis and kidney disease progression. Integrins are involved in podocyte attachment to the glomerular basement membrane. We hypothesized that in Fabry nephropathy, lyso-Gb3 could modulate αvβ3 expression in podocytes. Together with UPAR, the αvβ3 integrin is a key mechanism involved in podocyte detachment and podocyturia. Methods: In cultured human podocytes stimulated with lyso-Gb3, the mRNA expression of the *ITGAV* and *ITGB3* genes encoding integrins αv and β3, respectively, was analyzed by RT-qPCR. Results: In cultured human podocytes, lyso-Gb3 at concentrations encountered in the serum of Fabry patients increased *ITGAV* and *ITGB3* mRNA levels within 3 to 6 h. This pattern of gene expression is similar to that previously observed for *PLAUR (UPAR)* gene expression but is in contrast to the delayed (24 h) upregulation of other markers of podocyte stress and mediators of injury, such as CD80, TGFβ1, CD74, Notch1, and HES. Conclusions: Human podocyte stress in response to glycolipid overload in Fabry nephropathy, exemplified by lyso-Gb3, is characterized by an early increase in the expression of components of the αvβ3/UPAR system, which contrasts with the delayed rise in the expression of other mediators of podocyte injury. This suggests that the αvβ3/UPAR system may be a therapeutic target in Fabry nephropathy.

## 1. Introduction

Fabry disease is an X-linked lysosomal storage disease caused by mutations in the *GLA* gene encoding α-galactosidase A, which results in increased intracellular and circulating levels of enzyme substrates, such as globotriaosylceramide (Gb3), globotriaosylsphingosine (Lyso-Gb3), and galabiosylceramide [1]. The accumulation of these glycosphingolipids alters the morphology and function of cells in target tissues [2,3,4]. However, the precise mechanisms by which these metabolites lead to cellular dysfunction remain elusive. Mechanical overload or the interaction of intracellular glycosphingolipids with ion channels or transporters were initially suggested to potentially contribute to tissue damage [5,6]. However, more recently, the toxicity of soluble extracellular molecules, such as lyso-Gb3, has been emphasized. Kidney disease in the form of pathological albuminuria represents the first manifestation of target organ injury in patients with the classic (more severe) form of Fabry disease. Albuminuria then progresses to overt proteinuria, which may eventually become nephrotic-range proteinuria. This progression points to podocyte injury as a key driver of kidney disease in Fabry patients. Podocytes are terminally differentiated cells that do not proliferate Appendix A. Instead, they accumulate large amounts of glycosphingolipids and may release large amounts of more toxic metabolites such as lyso-Gb3, which may result in eventual detachment and excretion in urine [7,8,9]. Initially, neighboring podocytes may increase in size and cover the denuded glomerular basement membrane. However, this will increase their sensitivity to injury and eventually the podocyte mass will become critically low, leading to progressive glomerulosclerosis and nephron mass reduction [9]. Irreversible podocytopenia may underlie the observation that enzyme replacement therapy (ERT) may slow but not stop Fabry CKD progression to end-stage kidney disease, particularly if started at advanced stages of the disease [10,11].

We have recently reported pathological podocyturia in the early stages of Fabry nephropathy, when glomerular filtration rate (GFR) is still preserved and proteinuria is absent [12]. Integrin interaction with specific extracellular matrix ligands is key for podocyte attachment to the glomerular basement membrane [13]. The transmembrane receptor urokinase-type plasminogen activator receptor (uPAR), encoded by the *PLAUR* gene, interacts with podocyte integrin αvβ3, activating integrin-actin binding and podocyte contraction. However, persistent uPAR-integrin coupling leads to mechanical cellular stress and podocyte detachment [14,15]. In this study, we explored the regulation of αvβ3 expression by lyso-Gb in podocytes, as we observed how urinary excretion of uPAR+ podocytes increased in Fabry patients and how lyso-Gb3 increased uPAR expression in cultured human podocytes [16].

## 2. Methods

### 2.1. Cell Culture and Reagents

Human podocytes were from an immortalized podocyte cell line transfected with a temperature-sensitive SV40 gene construct and a gene encoding the catalytic domain of human telomeres, donated by Professor Moin Saleem (University of Bristol) [17,18,19]. At a permissive temperature of 33 °C, the cells remained in an undifferentiated proliferative state while at a 37 °C. These cells were growth arrested and differentiated to a podocyte phenotype. Undifferentiated podocytes were maintained at 33 °C in RPMI 1640 with penicillin, streptomycin, ITS (insulin, transferrin, selenite), and 10% FCS. At 70 to 80% confluence in 60-mm culture dishes, the temperature was increased to 37 °C for at least 14 days, until differentiation was confirmed [16,18,19,20,21,22,23,24,25]. The differentiation state was routinely assessed by well-known morphological changes corresponding to podocyte differentiation (Figure 1). Nondifferentiated podocytes showed typical epithelial cobblestone morphology and were small in size. In contrast, differentiated podocytes were characterized by enlarged cell bodies with an irregular shape with longer and spindle-like projections of arborizing podocytes. Podocyte cultures that failed to differentiate were discarded. Expression of podocyte markers such as nephrin, synaptopodin, or densin were periodically assessed to ensure correct identification [16,18,19,20,21,22,23,24,25]. Cells were cultured in serum-free media from 24 h before adding 100 nM of lyso-Gb3 (Sigma, St. Louis, MO, USA) and throughout the experiment. Lyso-Gb3 tested negative for lipopolysaccharide using the Limulus Amebocyte Lysate (LAL) Endotoxin Quantitation Kit (Termofisher, Waltham, MA, USA). This lyso-Gb3 concentration was clinically important, as circulating lyso-Gb3 concentration in classic Fabry disease ranges from 10 to 50 nM in females and >100 nM in males [26]. Prior dose-response studies in cultured podocytes had confirmed the bioactivity of this lyso-Gb3 concentration [18,19]. No changes in morphology or viability as a result of this exposure were observed.

### 2.2. Real-Time Reverse Transcription-Polymerase Chain Reaction (RT-PCR)

RNA was isolated using the Trizol reagent (Invitrogen, Paisley, UK) and 1 µg RNA was reverse transcribed using the High Capacity cDNA Archive Kit (Applied Biosystems, Foster City, CA, USA). Quantitative RT-PCR was performed on the ABI Prism 7500 sequence detection PCR system (Applied Biosystems, Foster City, CA) following the manufacturer’s protocol and the delta delta Ct method [18,19]. Gene expressions were provided as ratios to the housekeeping gene *GAPDH*. Pre-developed primer and probe assays were bought from Applied Biosystems. Three independent experiments were performed and every sample was assessed in duplicate, using the mean of two samples as the independent experiment data point.

### 2.3. Western Blot

Procedures for cell sample homogenization in lysis buffer, separation by 10% or 12% SDS-PAGE under reducing conditions and transfer to PVDF membranes (Millipore, Bedford, MA, USA) have been previously described [27]. Membranes were blocked with 5% skimmed milk in PBS/0.5% *v*/*v* Tween 20 for 1 h and washed with PBS/Tween. Primary antibodies were rabbit monoclonal anti-integrin alpha V (ITGAV, 1:500, Abcam, Cambridge, UK) and mouse monoclonal anti-integrin beta 3 (ITGB3, 1:500, Santa Cruz, CA, USA). Antibodies were diluted in 5% milk PBS/Tween. Blots were washed with PBS/Tween and then incubated with the respective horseradish peroxidase-conjugated secondary antibody (1:2000, GE Healthcare/Amersham, Aylesbury, UK). After washing, blots were developed with the chemiluminescence method (ECL) and re-probed with monoclonal anti-mouse α-tubulin antibody (1:2000, Sigma, St. Louis, MO, USA) to account for minor differences in loading.

### 2.4. Statistical Analysis

Results were expressed as mean ± SD. Differences between groups were analyzed using the Mann–Whitney test with Bonferroni correction. Correlations between variables were analyzed with Spearman’s rank correlation coefficient. Results were interpreted as significant if *p* < 0.05. The statistical program employed was InfoStat 2016, Córdoba, Argentina.

## 3. Results

### 3.1. Lyso-Gb3 Increases ITGAV and ITGB3 Gene Expression in Cultured Human Podocytes

uPAR+ podocyturia is a feature of early Fabry nephropathy, and lyso-Gb3 has been previously shown to increase the expression of mediators of kidney injury in podocytes, including UPAR, with peak response observed at 100 nM lyso-Gb3, a concentration found in the circulation of Fabry patients [17,18,19]. Thus, we explored the impact of lyso-Gb3 on the expression of the *ITGAV* and *ITGB3* genes, encoding integrins αv and β3, respectively, in cultured human podocytes. Lyso-Gb3 (100 nM) increased the mRNA expression of *ITGAV* mRNA that peaked at 3 h (Figure 1A). It also increased the mRNA expression of *ITGB3* mRNA, which had already been significant at 3 h and peaked at 6 h (Figure 1B).

### 3.2. Early Markers of Lyso-Gb3-Induced Stress in Cultured Human Podocytes

This pattern of early gene upregulation in response to lyso-Gb3 was previously observed for uPAR [16]. Indeed, there was a correlation between *ITGAV* and *uPAR* (urokinase-type plasminogen activator receptor) mRNA expression (Figure 2A), and there was also a trend toward correlation between *ITGB3* and *uPAR* mRNA expression (Figure 2B). This early upregulation pattern contrasted with the delayed onset of CD80, TGFβ1, CD74, Notch1, and HES upregulation [19,25], suggesting that changes in the expression of the UPAR/αvβ3 integrin system were one of the earliest manifestations of podocyte stress in response to lysoGb3 (Figure 3 and Appendix A).

### 3.3. Lyso-Gb3 Increases Integrin Av and Β3 Protein in Cultured Human Podocytes

Protein expression was evaluated by Western blot. Peak αV and β3 integrin protein levels were observed at 24 h (Figure 4), following the increase in mRNA expression, which is consistent with changes in gene expression driving changes in protein levels. As was the case for gene expression, the response was stronger for αV integrin than for β3 integrin.

## 4. Discussion

The key finding of this study is that the stress response of human podocytes to lyso-Gb3, a key glycosphingolipid accumulated in Fabry disease, is characterized by the early upregulation of genes encoding the αvβ3 integrin. Together with the early increase in *uPAR* gene expression in response to lyso-Gb3 [16], these data suggest a role of the UPAR/αvβ3 integrin system in podocyte injury during Fabry disease.

Fabry nephropathy is characterized by evidence of histological podocyte injury from childhood [28], which is followed by pathological albuminuria, also evidenced in childhood in males with classic Fabry disease. Proteinuria gradually increases to nephrotic levels, generally without nephrotic syndrome, as a manifestation of progressive glomerulosclerosis. As podocyte loss increases, patients with Fabry nephropathy will progress eventually to end-stage kidney disease. At this stage, they will require renal replacement therapy at the mean age of 40 years [10,29,30]. Proteinuria, as a marker of podocyte injury and loss, is a key prognostic factor for adverse kidney outcomes, both in untreated and in ERT-treated patients [31,32]. Podocyte loss is clinically interpreted as pathological podocyturia. In Fabry patients, pathological podocyturia can be observed when the urinary excretion of uPAR positive podocytes is increased [12,16].

Preservation of the podocyte population is critical for the function of the glomerular filtration barrier. Excessive urinary podocyte loss can precede the identification of pathological albuminuria due to disruption of the glomerular filtration barrier and the eventual decline in renal function due to nephron loss [9]. Despite ERT, Fabry patients remain at risk of progressive kidney function loss, particularly if significant podocyte loss has already taken place, as evidenced by the presence of proteinuria or glomerulosclerosis [10]. The rate of CKD progression and response to therapy may be influenced by many factors, including genetic background (in particular, the severity of *GLA* mutation), comorbidities, age at ERT initiation, and ERT dosage. In index cases, ERT is usually initiated later in the course of the disease [7,10,12]. In this regard, early therapeutic intervention may prevent irreversible podocyte detachment and loss [8]. Identification of the molecular mechanisms that facilitate podocyte detachment in response to glycolipid accumulation is key to designing novel therapeutic approaches that preserve podocyte numbers in Fabry nephropathy.

Integrins are key modulators of podocyte adhesion to the glomerular basement membrane. They are heterodimeric transmembrane proteins containing one α and one β subunit. Integrins mediate cell-cell and cell-matrix interactions [8,33]. Disruption of integrin distribution and/or activity in podocytes and/or integrin ligands in the glomerular basement membrane may both reflect and cause podocyte stress and injury [8,34]. Thus, integrin disruption may facilitate podocyte detachment and podocyturia, leading to proteinuria and progressive loss of kidney function. Integrin αVβ3 is the vitronectin receptor. In Fabry patients, urinary integrin αVβ3 and podocyte β3 expression were reported to be increased, suggesting the involvement of αVβ3 in the pathogenesis of Fabry podocytopathy [35]. uPAR is a glycan phosphatidylinositol (GPI)-linked protein that can associate with αVβ3 to regulate cell migration, tumor invasion, and host defense [36,37]. The complexing of lipid raft-associated uPAR with podocyte αVβ3 modulates uPAR’s ligand-binding activities [14,38,39]. Specifically, uPAR binding to β3 integrin results in podocyte integrin activation, cell contraction, eventual podocyte detachment and podocyturia, potentially contributing to podocyte loss [12,15,35]. The present and recent data from our group demonstrates that αVβ3 integrin and uPAR expression are early markers of podocyte stress induced by metabolites accumulated in Fabry disease, specifically by lyso-Gb3. Moreover, the finding of a coordinated upregulation of the expression of different components of the system suggest a relationship with podocyte injury and potentially podocyturia in Fabry disease. In this regard, uPAR was expressed through podocyte detachment in the urine of Fabry patients. uPAR was also expressed through the antiproteinuric effect of amiloride in rat proteinuria, which was related to reduced uPAR podocyte expression and podocyte motility [40]. Interestingly, podocyturia decreased in one Fabry patient treated with amiloride [41].

Lyso-Gb3 is a promising biomarker in Fabry disease which may also contribute to the pathogenesis of target organ injury. Patients with milder *GLA* mutations have late onset Fabry disease and circulating lyso-Gb3 around 3−18 nM, while severe *GLA* mutations causing classical Fabry disease and nephropathy display lyso-Gb3 levels in the 80−300 nM range in males [20,42]. This latter range is the one tested in the present study. Interestingly, late-onset mutations with low lyso-Gb3 levels rarely cause end-stage kidney disease, despite the presence of abundant podocyte glycolipid deposits [43,44], further suggesting a potential role for lyso-Gb3 in Fabry nephropathy progression.

The precise early intracellular mediators of lyso-Gb3 biological responses in podocytes and other cell types are unclear. In cultured human podocytes, lyso-Gb3 activated Notch1 signaling, resulting in increased active Notch1 and HES1, a canonical Notch transcriptional target [18]. Notch1 siRNA or γ-secretase inhibition also prevented the lyso-Gb3-induced upregulation of Notch1, Notch ligand Jagged1, and chemokine (MCP1, RANTES) expression, suggesting that activation of baseline expressed Notch1 may be an early signaling pathway. Notch activation is triggered by ligand binding leading to proteolytic cleavage and release of the Notch intracellular domain, which enters the cell nucleus to behave as a transcription factor and regulate gene expression [45]. Nuclear factor kappa B (NFκB) is another transcription factor engaged by lyso-Gb3 [18]. Likely, delayed gene expression, which peaked at around 24 h in response to lyso-Gb3, is driven by the early engagement of Notch with NFκB. Thus, the two-peak (3 h and 24 h) chemokine response to lyso-Gb3 in podocytes likely depends on two waves of NFκB activation, with the second wave dependent on Notch1 signaling, which also mediates fibrogenic responses, as suggested by inhibition by Notch siRNA or parthenolide [18]. In summary, the available information suggests an early activation of baseline expressed Notch1 and of NFκB in response to lyso-Gb3 stimulation in podocytes and a second wave of NFκB activation mediated by Notch1 and potentially other drivers. In this regard, NFκB has long been known to drive uPAR expression [46]. Interestingly, uPAR expression induced by the phospholipid lysophosphatidic acid also required NFκB [47]. More recently, Notch1 was identified as a driver of uPAR expression [48]. Additionally, uPAR may contribute to Notch1 activation, effectively closing a positive feedback loop [49]. The αvβ3 integrin is also a known activator of NFκB [50], potentially contributing to the second wave of NFκB activation induced by lyso-Gb3 in podocytes. However, there is less information on the interaction of Notch1 and the αvβ3 integrin.

Certain limitations should be acknowledged. The cell culture system only allowed acute experiments, and it was not feasible to reproduce the chronic (years’ worth of) exposure of podocytes from Fabry patients to lyso-Gb3. Thus, while the system allowed us to illustrate a biological activity over mediators of podocyte injury in cultured huma podocytes, it did not allow us to track the long-term response. It also did not allow us to track the reversibility of the changes that occur when lyso-Gb3 is no longer present in the cell culture media and simulating the in vivo response to Fabry therapy. Even if long-term exposure (e.g., weeks) were possible, this would, in any case, take place in a cell culture environment in the absence of endothelial cells. However, in contrast to other approaches, including in vivo studies in Fabry mice, this system allows for the exploration of responses in human cells. Additionally, this system allows us to dissociate the direct impact of lyso-Gb3, which remains high despite ERT or chaperone therapy in humans in vivo, from the impact of glycolipid deposits, which may be cleared by high doses of ERT [51].

## 5. Conclusions

Podocyte stress in response to bioactive glycolipids accumulated in Fabry disease, such as lyso-Gb3, is characterized by an early increase in the gene expression of components of the uPAR/αvβ3 integrin system, which may contribute to podocyte detachment and podocyturia. Therapeutic targeting of this system should be explored as an adjuvant therapy for Fabry nephropathy, given that ERT decreases but does not normalize lyso-Gb3 levels.

## Figures and Tables

**Figure 1 jcm-09-03659-f001:**
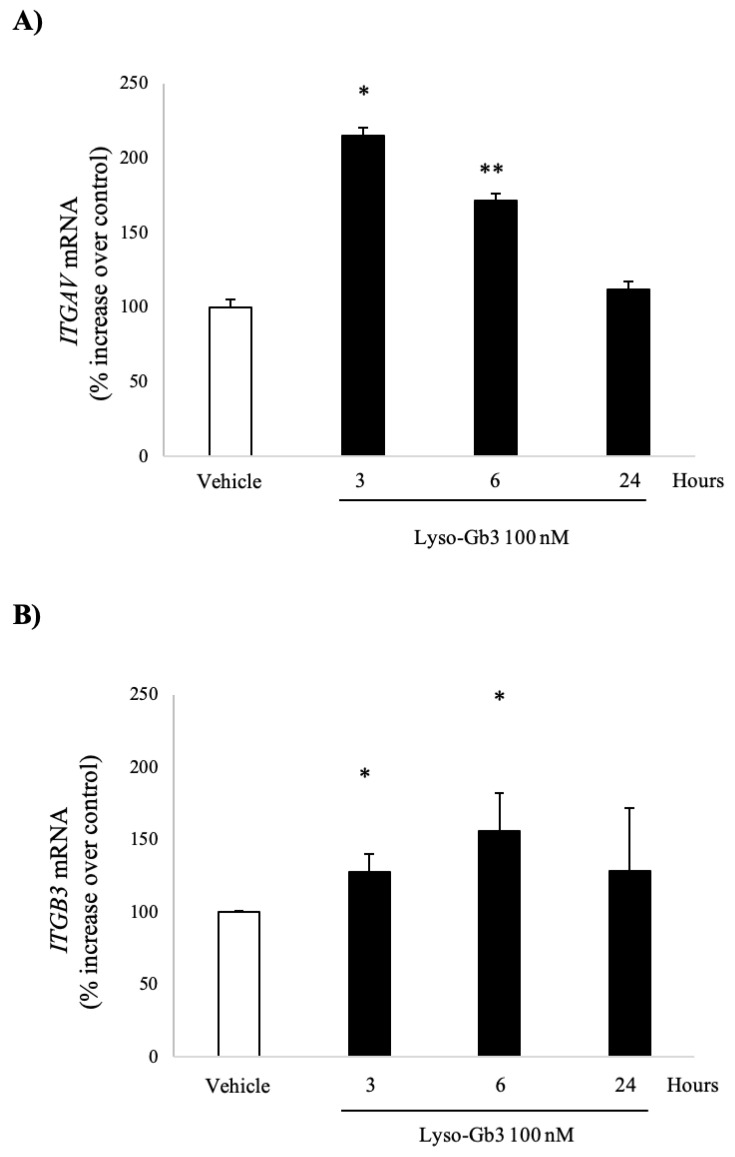
Globotriaosylsphingosine (Lyso-Gb3) upregulates *ITGAV* and *ITGB3* expression in human podocytes. *ITGAV* and *ITGB3* encode integrins αv and β3, respectively. Cultured human podocytes were stimulated with 100 nM lyso-Gb3. (**A**) Time-course of *ITGAV* mRNA expression in response to lyso-Gb3. * *p* < 0.01 vs. control, ** *p* < 0.02 vs. control. (**B**) Time-course of *ITGB3* mRNA expression in response to lyso-Gb3. * *p* < 0.02 vs. control. Expression of mRNA was assessed by real-time RT-qPCR. Mean ± SEM (standard error of the mean) of three independent experiments.

**Figure 2 jcm-09-03659-f002:**
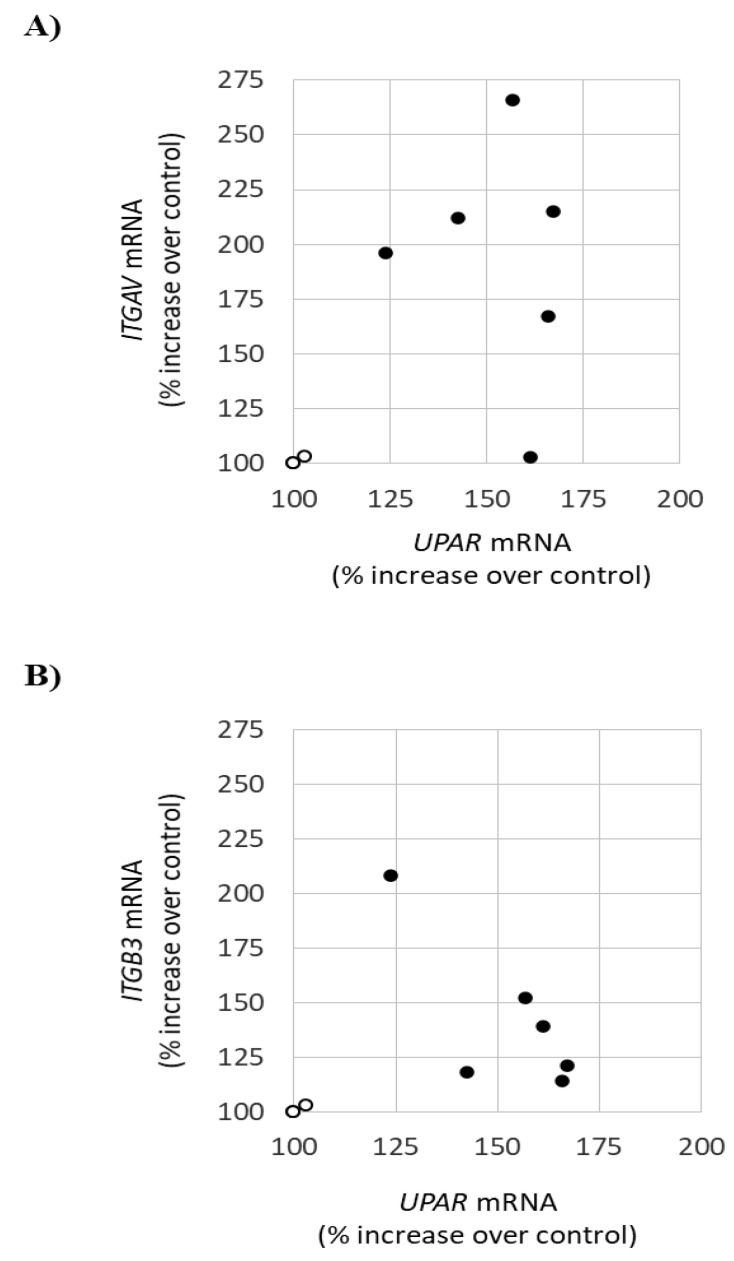
Relationship between expression of integrin genes and *uPAR (PLAUR)* gene expression in cultured podocytes in response to lyso-Gb3. Cultured human podocytes were stimulated with 100 nM lyso-Gb3. (**A**) Correlation between *ITGAV* and *UPAR* mRNA levels. R^2^ 0.45, *p* = 0.049. (**B**) Correlation between *ITGB3* and *uPAR* mRNA levels. R^2^ 0.30, *p* = 0.13, ns. Expression of mRNA was assessed by real-time RT-qPCR. Open circles represent values in control podocytes, black circles represent values in lyso-Gb3 treated samples. Please note overlap between two control data points. Data points at 3 and 6 h were included in the analysis, since by 24 h gene expression is already decreasing. uPAR data have been previously published [16].

**Figure 3 jcm-09-03659-f003:**
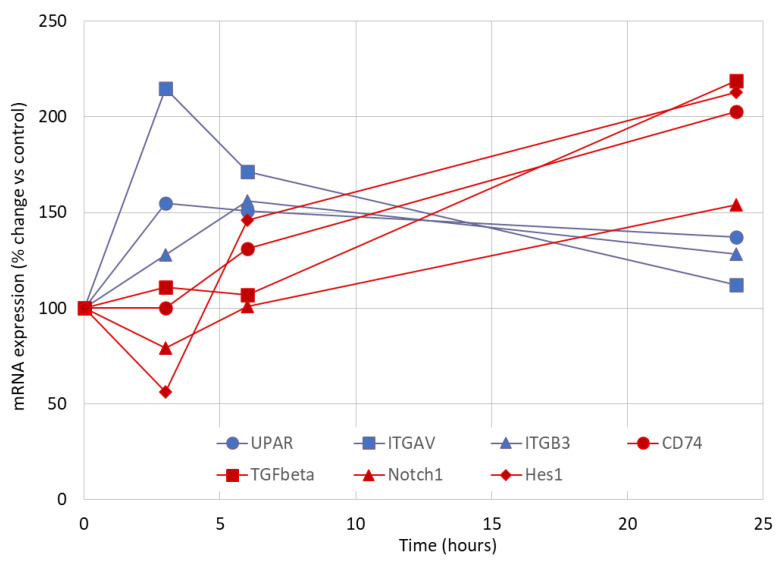
Gene expression stress response to lyso-Gb3 in cultured human podocytes. The stress response to lyso-Gb3 was characterized by the early increase in genes encoding proteins of the integrin/UPAR system (*ITGAV*, *ITGB3*, and *UPAR*, blue data points) and a delayed increase in the expression of genes encoding additional mediators of inflammation and fibrosis (CD80, TGFβ1, CD74, Notch1, and HES, red data points). Data from the present publication and from [16,18,19,25], in which the same podocytes were tested. Appendix A
Appendix A represents the figure with error bars. Cluster Differentiation 80 (CD80), Transforming growth factor β1 (TGFβ1), Cluster Differentiation 74 (CD74), Notch-1 (Notch-homolog-1), Hairy and enhancer of split-1 (HES).

**Figure 4 jcm-09-03659-f004:**
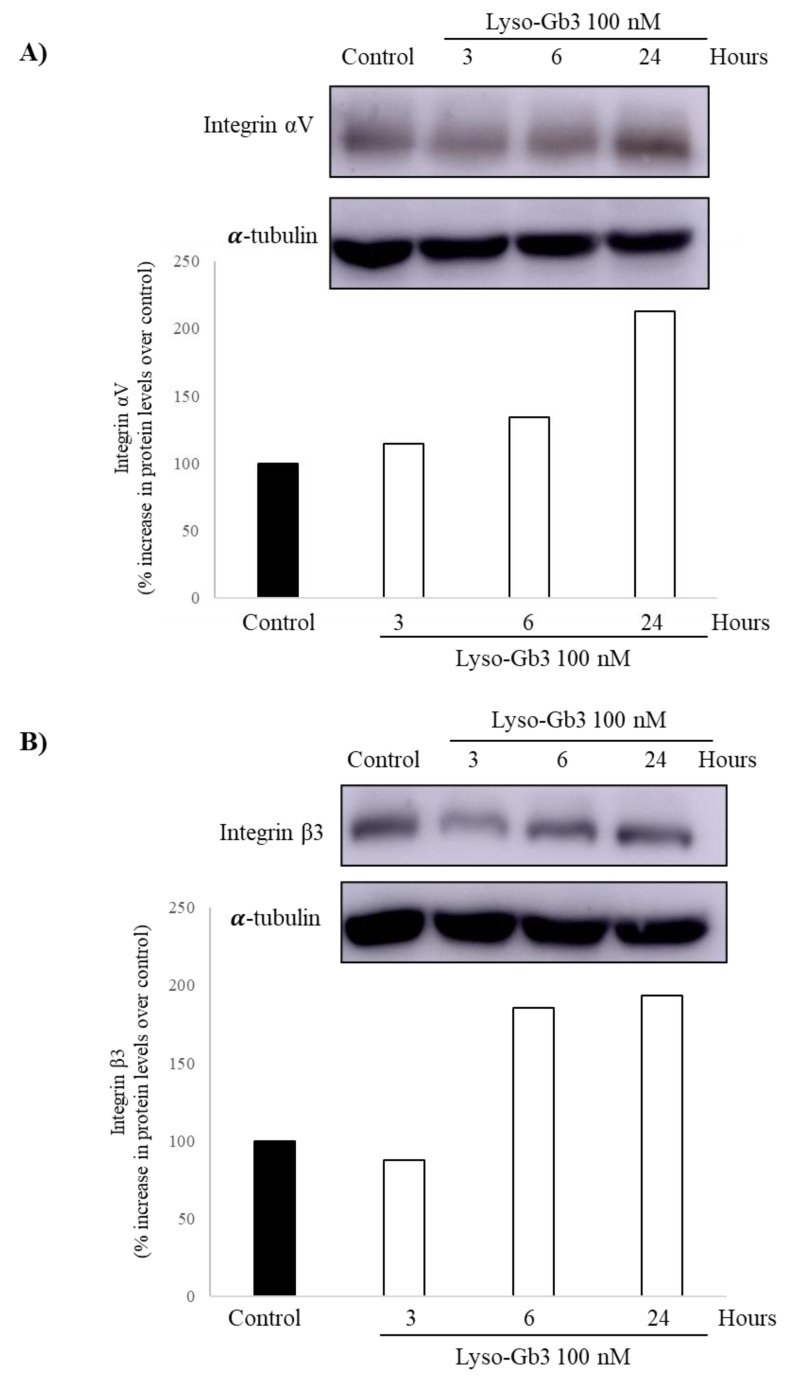
Lyso-Gb3 and integrin expression in cultured human podocytes. Integrin expression was assessed by Western blot in samples from podocytes cultured in presence of 100 nM lyso-Gb3. (**A**) Integrin αV, encoded by *ITGAV*. (**B**) Integrin β3, encoded by *ITGB3*.

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
