# Peer review of "Lyso-Gb3 Increases αvβ3 Integrin Gene Expression in Cultured Human Podocytes in Fabry Nephropathy"

_jcm, 2020, doi:10.3390/jcm9113659_

Round 1

Reviewer 1 Report

Fabry is a genetic disease that can affect a person for their entire lives and would presumably lead to raised levels Lyso-Gb3 for years. So how relevant is the first few hours of response to Lyso-Gb3? Of what significance is knowing the difference in RNA expression between hours 3 and 6? Podocytes have a slow turnover time, so it seems like tracking the response over long Lyso-Gb3 exposures (on the order of days to weeks) would be more useful in understanding the podocyte response to this clinical condition. Is the subsequent protein upregulation sustained for longer than 24 hrs? Does the RNA and protein recover back to normal levels if the Lyso-Gb3 is removed, which might indicate that medical intervention could reverse the damage done during the course of the disease. Do the cells changes morphology or viability as a result of this exposure? The scope of this paper is very limited and it seems like in its current form the paper doesn’t explore the connection between podocyte physiology and lysoGb3 well enough to begin to justify drawing clinically relevant conclusions.

Additional comments:

What plate/well size were the cells cultured in?

It says the cells were grown for at least 14 days until differentiation was confirmed. How was differentiation confirmed? Did you do it for all cells used in this paper? Those results should be shown to confirm differentiated adult podocytes were achieved.

What does ‘Lyso-Gb3 tested negative for lipopolysaccharide’ mean? Did you test it and how.

How many samples were taken for the RT PCR experiments? And were they from technical or biological replicates? In the caption for figure 1 it says ‘data is from 3 independent experiments’. That should be clarified in the methods section.

The Mann-Whitney test is used for comparing just 2 groups, but in your data you are comparing more than 2 (control, 3, 6, 24 hrs). So a multiple comparison adjustment should be made or something like an ANOVA with multiple comparisons should be used.

In Fig2 the caption does not match the figure. There are no open circles or black squares.

Figure 3: the x-axis needs to be labeled so it’s obvious what the units are. I would suggest just connecting the points with straight lines, unless there’s a justifiable reason for using some sort of polynomial fit. The error bars should be included. Without the error bars to show how much variability there is within the samples it’s difficult to interpret how meaningful this data is.

Are there error bars for fig 4?

Author Response

Reviewer 1

Fabry is a genetic disease that can affect a person for their entire lives and would presumably lead to raised levels Lyso-Gb3 for years. So how relevant is the first few hours of response to Lyso-Gb3? Of what significance is knowing the difference in RNA expression between hours 3 and 6? Podocytes have a slow turnover time, so it seems like tracking the response over long Lyso-Gb3 exposures (on the order of days to weeks) would be more useful in understanding the podocyte response to this clinical condition. Is the subsequent protein upregulation sustained for longer than 24 hrs? Does the RNA and protein recover back to normal levels if the Lyso-Gb3 is removed, which might indicate that medical intervention could reverse the damage done during the course of the disease. Do the cells change morphology or viability as a result of this exposure? The scope of this paper is very limited and it seems like in its current form the paper doesn’t explore the connection between podocyte physiology and lysoGb3 well enough to begin to justify drawing clinically relevant conclusions.

R: The reviewer raised a series of valid points on the clinical significance of the findings. These are now discussed in the Discussion section in a new limitations paragraph as follows:

“Certain limitations should be acknowledged. The cell culture system only allowed acute experiments and was not feasible to reproduce the chronic (years) exposure of podocytes from Fabry patients to lyso-Gb3. Thus, while the system allows to illustrate a biological activity over mediators of podocyte injury in cultured huma podocytes, it did not allow to track the long term response and the reversibility of the changes when lyso-Gb3 is no longer present in the cell culture media simulating the in vivo response to Fabry therapy. Even if long term exposure (e.g. weeks) were possible, this would in any case take place in a cell culture environment in the absence of endothelial cells. However, by contrast to other approaches, including in vivo studies in Fabry mice, this system allows the exploration of responses in human cells and additionally allows to dissociate the direct impact of lyso-Gb3, which remains high despite ERT or chaperon therapy in humans in vivo, from the impact of glycolipid deposits, which may be cleared by high doses of ERT”

Do the cells changes morphology or viability as a result of this exposure?”

R: We thank the reviewer for this relevant question. We did not observe such changes as currently indicated in the text.

Additional comments:

  1. What plate/well size were the cells cultured in?

R: Cells were cultured in 60 mm culture dishes, as now indicated in the text

  1. It says the cells were grown for at least 14 days until differentiation was confirmed. How was differentiation confirmed? Did you do it for all cells used in this paper? Those results should be shown to confirm differentiated adult podocytes were achieved.

R: The differentiation state is routinely assess by the well-known morphological changes corresponding to podocyte differentiation (Supplementary figure 1). Nondifferentiated podocytes showed typical epithelial cobblestone morphology and are small in size. In contrast, differentiated podocytes are characterized by enlarged cell bodies with an irregular shape with longer and spindle like projections of arborizing podocytes. Podocyte cultures that fail to differentiate are discarded. Expression of podocyte markers such as nephrin, synaptopodin or densin are periodically assessed to ensure their identity. This is now indicated in the manuscript and a suppl figure and further references have been added.

  1. What does ‘Lyso-Gb3 tested negative for lipopolysaccharide’ mean? Did you test it and how.

R: The Limulus Amebocyte Lysate (LAL) Endotoxin Quantitation Kit was used, as now indicated in methods.

  1. How many samples were taken for the RT PCR experiments? And were they from technical or biological replicates? In the caption for figure 1 it says ‘data is from 3 independent experiments’. That should be clarified in the methods section.

R: We apologies for the lack of clarity. This is now explained as follows: Three independent experiments were performed and every sample was assessed in duplicate, using the mean of two samples as the independent experiment data point.

  1. The Mann-Whitney test is used for comparing just 2 groups, but in your data you are comparing more than 2 (control, 3, 6, 24 hrs). So a multiple comparison adjustment should be made or something like an ANOVA with multiple comparisons should be used.

R: Apologies, the Bonferroni correction option is on when performing the analysis. This is now stated in the manuscript.

  1. In Fig2 the caption does not match the figure. There are no open circles or black squares.
  2. Apologies, we redraw the figure to fit with the overall style and this these dots were not changed. This has now been corrected.
  3. Figure 3: the x-axis needs to be labeled so it’s obvious what the units are. I would suggest just connecting the points with straight lines, unless there’s a justifiable reason for using some sort of polynomial fit. The error bars should be included. Without the error bars to show how much variability there is within the samples it’s difficult to interpret how meaningful this data is.

R: The figure has now been changed as suggested. Given that the error bars may overpopulate the figure, we have drawn a new suppl figure that represents figure 3 with error bars. We apologize for missing the x axis label, this was a copy-paste error from ppt, where the figure was build, to the word file that was submitted. This has now been incorporated to the figure.

  1. Are there error bars for fig 4?

R: Figure 4 has no error bars

Reviewer 2 Report

Although I find it as an interesting paper showing the αvβ3 integrin expressed in early stage of podocyte injury in Lyso-Gb3 treated podocytes, the paper would benefit from the following improvements.

> Discussion
1) There seems to be a link between uPAR and Integrin. Are there any specific pathways that can explain this?
2) What is the possible explanation for the earlier expression of ITGAV and ITGB3 compared to other mediators (TGF beta, Notch1, Hes1, CD74)?
Please add it to the discussion section.

> Minor comments
: Several text editing is needed (line 31, line 84, line 159,...)
: Figure 3. Does the X axis mean hours?

Author Response

Reviewer 2

Although I find it as an interesting paper showing the αvβ3 integrin expressed in early stage of podocyte injury in Lyso-Gb3 treated podocytes, the paper would benefit from the following improvements.

Discussion

1) There seems to be a link between uPAR and Integrin. Are there any specific pathways that can explain this?

2) What is the possible explanation for the increase in lyso-Gb3 in the early stages of podocyte injury compared to other mediators (TGF beta, Notch1, Hes1, CD74)?

Please add it to the discussion section.

R: Point 1) and 2) are now addressed in the Discussion as follows: “The precise early intracellular mediators of lyso-Gb3 biological responses in podocytes and other cell types are unclear. In cultured human podocytes, lyso-Gb3 activated Notch1 signaling, resulting in increased active Notch1 and HES1, a canonical Notch transcriptional target. Notch1 siRNA or γ-secretase inhibition also prevented the lyso-Gb3-induced upregulation of Notch1, Notch ligand Jagged1 and chemokine (MCP1, RANTES) expression, suggesting that activation of baseline expressed Notch1 may be an early signaling pathway. Notch activation is triggered by ligand binding leading to proteolytic cleavage and release of the Notch intracellular domain, which enters the cell nucleus to behave as a transcription factor and regulate gene expression (ref). Nuclear factor kappa B (NFκB) is another transcription factor engaged by lyso-Gb3. Likely, delayed gene expression, peaking at around 24h, in response to lyso-Gb3 is driven by the early engagement of Notch and NFκB. Thus, the two-peak (3h and 24h) chemokine response to lyso-Gb3 in podocytes likely depends on two waves of NFκB activation, the second one dependent on Notch1 signaling, which also mediates fibrogenic responses, as suggested by inhibition by Notch siRNA or parthenolide. Thus, available information suggests early activation of baseline expressed Notch1 and of NFκB in response to lyso-Gb3 stimulation in podocytes and a second wave of NFκB activation mediated by Notch1 and potentially other drivers. In this regard, NFκB has long been known drive uPAR expression (ref). Interestingly, uPAR expression induced by the phospholipid lysophosphatidic acid also required NFκB (ref). More recently, Notch1 was identified as a driver of uPAR expression (ref). Additionally, uPAR may contribute to Notch1 activation, effectively closing a positive feed-back loop (ref). The αvβ3 integrin is also a known activator of NFκB (ref), potentially contributing to the second wave of NFκB activation induced by lyso-Gb3 in podocytes. However, there is less information on the interaction of Notch1 and the αvβ3 integrin.”

Minor comments

Several text editing is needed (line 31, line 84, line 159,...)

               R: Thank you for pointing this out. This has now been corrected. Other typos were also corrected.

Figure 3. Is Y axis hours?

R: Thank you for pointing this out. This has now been corrected.

Round 2

Reviewer 1 Report

The author's responses and changes were sufficient to address my previous comments.

This manuscript is a resubmission of an earlier submission. The following is a list of the peer review reports and author responses from that submission.